# Mathematical Model for the Biological Control of the Coffee Berry Borer *Hypothenemus hampei* through Ant Predation

**DOI:** 10.3390/insects14080675

**Published:** 2023-07-29

**Authors:** Carlos Andrés Trujillo-Salazar, Gerard Olivar-Tost, Deissy Milena Sotelo-Castelblanco

**Affiliations:** 1Faculty of Basic Sciences, Universidad del Quindío, Armenia 630001, Colombia; catrujillo@uniquindio.edu.co; 2Department of Mathematics and Statistics, Universidad Nacional de Colombia, Manizales 170004, Colombia; dmsoteloca@unal.edu.co; 3Department of Natural Sciences and Technology, Universidad de Aysén, Coyhaique 5950000, Chile

**Keywords:** predator–prey model, coffee berry borer, ants, biological control, net reproductive rate, nonhyperbolic equilibrium points, transcritical bifurcation

## Abstract

**Simple Summary:**

Coffee is an essential agricultural product in the worldwide economy. Unfortunately, the coffee berry borer (CBB), the crop’s primary pest, significantly impacts the bean’s quantity and quality. One strategy for controlling this insect is to use biological control agents, such as ants, some known for naturally living on coffee plantations and consuming CBB throughout all stages of development. Our paper examines a predator–prey interaction between these two insects using a novel mathematical model based on differential equations, where state variables correspond to adult CBBs, immature CBBs, and ants.

**Abstract:**

Coffee is a relevant agricultural product in the global economy, with the amount and quality of the bean being seriously affected by the coffee berry borer *Hypothenemus hampei* (Ferrari), CBB, its principal pest. One of the ways to deal with this beetle is through biological control agents, like ants (Hymenoptera: Formicidae), some of which are characterized by naturally inhabiting coffee plantations and feeding on CBB in all their life stages. Our paper considers a predator–prey interaction between these two insects through a novel mathematical model based on ordinary differential equations, where the state variables correspond to adult CBBs, immature CBBs, and ants from one species, without specifying whether preying on the CBB is among their feeding habits, in both adult and immature stages. Through this new mathematical model, we could qualitatively predict the different dynamics present in the system as some meaningful parameters were varied, filling the existing gap in the literature and envisioning ways to manage pests. Mathematically, the system’s equilibrium points were determined, and its stability was studied through qualitative theory. Bifurcation theory and numerical simulations were applied to illustrate the stability of the results, which were interpreted as conditions of the coexistence of the species, as well as conditions for eradicating the pest, at least theoretically, through biocontrol action in combination with other actions focused on eliminating only adult CBBs.

## 1. Introduction

The coffee berry borer (CBB), *Hypothenemus hampei* (Ferrari, 1867) (Coleoptera: Curculionidae: Scolytidae), has become a critically important pest in the coffee industry worldwide. Its devastating impact on coffee production and quality has attracted the attention of farmers, scientists, and decision-makers. This tiny beetle, considered the primary coffee pest [1,2,3], is the size of the head of a pin and penetrates the beans by reproducing inside them, directly affecting the yield and quality of the crop. Furthermore, the CBB represents a significant economic and social threat, as it reduces farmers’ incomes, decreases the competitiveness of coffee-growing regions, and affects communities dependent on this critical industry.

The CBB destruction cycle begins when an adult female CBB enters through the navel of a coffee bean and, once in the endosperm, makes galleries where it deposits its eggs. These hatches give rise to two other immature stages: larva and pupa, with the larval stage affecting the grain the most. First, CBBs emerge from the pupae in the adult state, in an approximate ratio of 1:10 in favor of females to males. Then, sibling CBBs mate (a phenomenon known as inbreeding), and the fertilized female CBBs leave the grain to continue the cycle. Some of these new adult female CBBs oviposit in the exact grain where they hatched but to a lesser extent than the founder borer. The males, for their part, remain inside the grain until they die, fulfilling an exclusively reproductive function since their wings and jaws are atrophied, preventing them from flying and perforating grains [4,5,6].

The ability of CBBs to develop within the coffee bean makes it challenging to control this pest. The protection the grain provides to the immature stages of CBBs (egg, larva, prepupa, pupa) makes detecting and applying effective control measures difficult. In addition, the ability of adult females to enter other grains to oviposit increases the spread of the pest within the crop [2,4,5]. Only through a comprehensive approach, combining scientific research and the development of effective control strategies, can this challenge be met and a sustainable future for the coffee industry ensured. Some of these control techniques are described below. (1) Chemical control consists of using insecticides to eliminate adult CBBs found outside the bean and which are prepared to infest another bean [5,7]. (2) Cultural control aims to minimize the pest’s availability of food and shelter by implementing distinct manual practices, such as using handmade traps with alcoholic attractants [3,5]. (3) Biological control, based on diminishing the effects of the CBB through the introduction into coffee plantations of other animals, is classified into four groups: parasitoids, predators, entomopathogens, and entomonematodes [5,8]. Finally, (4) integrated borer management (IBM), corresponding to a strategic combination of chemical, cultural, and biological controls, granting preference to the latter, aims to diminish the CBB population to levels that do not cause significant damage to coffee production while protecting the crops and minimizing the damage caused to the environment [3,5,8].

Biological control is very relevant in the fight against the CBB, not only due to its importance within IBM but because it can be carried out with a large variety of living beings, among which predators stand out, constituting the largest group of natural enemies of the CBB. One predator that deserves special attention is the ant (Hymenoptera: Formicidae). Ants are very effective in CBB control due to their size and the ease of introducing them into the affected bean, where they find CBB available in all its life stages [2,5]. However, once the ant enters the coffee bean, it does not necessarily prey on all stages; some prefer adult CBBs, like, for example, the species *Solenopsis picea*, according to [9], or the species *Pseudomyrmex simplex*, *Pseudomyrmex ejectus*, and *Pseudomyrmex* sp., according to [10]. In addition, ants from the species *Crematogaster* sp. contribute to biological control by consuming immature CBBs inside the coffee beans [11]. Other ants, which cannot enter the beans, concentrate their predation on adult CBBs that have exposed >50% of their body when they are perforating the fruit or simply on adult CBBs that walk among beans, on the branches of the coffee plants, or on the soil, such as the species *Solenopsis invicta* and *Gnamptogenys annulata*, according to [12,13], respectively. Other studies have revealed that certain ant species that feed on CBB attack this pest in all its life stages, like the species *Solenopsis geminata* [13], without this being the only case. According to [14], the species *Wasmannia auropunctata* preys on immature CBBs inside the coffee beans, but [12], a more recent study, stated that this same species consumes adult CBBs, also inside the beans. According to these reports, it may be said that ants of the species *Wasmannia auropunctata* have in their diet immature CBBs and adult CBBs.

Bearing in mind the aforementioned details, this work proposed and analyzed a mathematical model comprising three ordinary differential equations to theoretically study the interaction between the CBB and ants that prey on the CBB as part of their feeding habits, in the adult and immature stages. Each of the equations is described according to the characteristics of the three populations in question: adult CBBs, immature CBBs, and CBB-preying ants. One of the purposes of this study was to interpret the stability of the results as conditions of species coexistence, as well as conditions to eradicate the pest, at least theoretically, by biocontrol action in combination with other actions focused on eliminating only adult CBBs; in particular, the influence of the death rate of adult CBBs due to factors other than predation on the population dynamics was studied. Based on these considerations, we wished to answer two specific questions: (1) Under what conditions could the population of CBBs coexist with the population of ants despite their role as predators? (2) Under what conditions might the CBB population be eradicated by ant biological control combined with other control methods? In other words, we desired to analyze, using a mathematical model, the reduction and hypothetical elimination of CBBs when biological control based on ants was implemented and the additional death factor of adult CBBs was systematically varied. In this sense, our study is novel in that we introduced both ant biological control and other mortality factors for CBBs, unlike works such as [15,16], where the CBB population dynamics were also modeled using systems of ordinary differential equations. The first of these references studied the effects of a biological control consisting of an entomopathogenic fungus. The second analyzed the synergy of chemical and cultural controls, specifically bioinsecticides and capture traps.

From the proposed mathematical model, a region of biological sense was deduced and shown to be positively invariant, that is, a set wherein the solutions for non-negative values of the independent variable time remained. Next, the equilibrium points of the system of differential equations were determined, and its stability was studied through the qualitative theory for continuous dynamical systems, that is, from the sign of the eigenvalues of the Jacobian matrix evaluated in each equilibrium, a sign that in some cases depended on one of the two thresholds found, referred to in the population ecology literature as net reproduction rate. Some of the stability results are discussed according to bifurcation theory. Thereafter, numerical simulations of the system of differential equations were carried out, and the respective interpretations are provided. Finally, the benefits of the work carried out and the results obtained are discussed.

## 2. Materials and Methods

This section proposes a mathematical model constituting three ordinary differential equations to study a prey–predator interaction, with CBBs and ants playing the respective roles. Each of the equations is described according to the characteristics of the three populations in question: adult CBBs, immature CBBs, and ants whose feeding habits include CBB predation.

An initial assumption was that the CBBs are divided into two populations: adult and immature, although ecologically they continue to be part of the same population. In addition, no sex distinction was made. The work also considered a population of ants that consume CBBs as part of their diet, notwithstanding the development stage. Other typical assumptions regarding the ecology of populations were considered in this model, as follows:The generations of the population overlapped, that is, in the population it was possible to find individuals of different ages.The population was homogeneously distributed, which is why position was irrelevant.No migration phenomena were considered.

Let A=A(t) designate the average number of adult CBBs at time *t*, I=I(t) the average number of immature CBBs at time *t*, and H=H(t) the average number of ants at time *t*. Thus, derivates $\frac{dA}{dt},\frac{dI}{dt}$, and $\frac{dH}{dt}$ represent the variation of each population with respect to time.

The model’s approach started by formulating a differential equation for the variation of the average number of adult CBBs; for this, three aspects that influence the dynamics were taken into account. In the first place, we considered the transition from immature stages to adult stages. If we denote with ω the rate of development from the immature stage to the adult stage, the term ωI corresponds to the average number of immature CBBs undergoing this transition per unit of time. In the second place, we considered the death of adult CBBs, with parameter μ assumed as the mortality rate of adult CBBs due to factors other than predation, such as natural death or death due to the effect of insecticides; thus, the term μA represents the average number of adult CBBs that die per unit of time. Third, we considered predation by ants from a certain species, which, without being explicitly identified, could be any of those mentioned in the introduction, as long as it feeds on the CBB at any moment of its life stage as part of its feeding habits. This interaction could be described by the law of mass action, a typical consideration in predator–prey-type models, which in this case is represented by the term αAH, where parameter α corresponds to the death rate due to the predation of adult CBBs by ants. According to the above, the population dynamics of adult CBBs was modeled by the following differential equation:(1)$$\frac{dA}{dt}=\omega I-\mu A-\alpha AH.$$

Now, a differential equation was proposed that modeled the variation in the average number of immature CBBs. The starting point was a common fact explained in the literature: that the product of the rate at which the eggs are produced and the variable corresponding to the population of adult CBBs describes the growth in the population of immature CBBs; in this case, said product is represented by ϕA, where ϕ is the CBB oviposition rate. However, immature CBBs do not increase in number indefinitely; this is because they have a carrying capacity, denoted in this case by *q* and defined in this context as the maximum number of immature CBBs that the cultivation area can withstand, a number that varies according to the plantation’s characteristics. It is important to recall that immature CBBs always remain in the coffee beans. When the average number of immature CBBs is close to the carrying capacity, the dynamic is stabilized, that is, growth is prevented. To model this situation, we multiplied the term ϕA by the factor $F=\left(1-I/q\right)$, which evidently lacks units, because these were simplified in the quotient. The regulatory effect of this factor is clear; when the number of immature CBBs is very close to the carrying capacity, that is, when I→q, then F→0, and, hence, growth is inhibited. This regulation factor has been used in different contexts, such as population dynamics studies of the dengue-transmitting mosquito *Aedes aegypti*, as in [17,18], and a demographic structure study of the plant *Emilia sonchifolia*, as in [19]. Moreover, the mortality rate of immature CBBs due to environmental factors is denoted by the parameter θ; then, the term θI represents the average number of immature CBBs that die per unit of time. It is worth mentioning that insecticides have a significantly negative effect on the population of CBBs inside the beans [2,5]. Lastly, the law of mass action was once again taken into account to describe the predation of immature CBBs *I* at a rate δ by the same species of ants *H* considered in the prior differential equation, a dynamic summarized in the term δIH. In sum, the population dynamics of the immature CBBs were modeled by the following differential equation:(2)$$\frac{dI}{dt}=\phi A\left(1-\frac{I}{q}\right)-\omega I-\theta I-\delta IH.$$

Finally, for the population dynamics of the average number of ants that prey on CBBs at both the adult and immature stages, logistical growth was considered, where *k* and *r* denote the carrying capacity and the intrinsic growth rate, respectively. This assumption was admissible given that in the CBBs’ absence, the ants are not extinguished, because they have other sources of food in the coffee plantations. Following the methodology proposed in [18,20,21,22], an additional increase was considered for the intrinsic growth of the ants attributed to the predation of adult CBBs εαA and immature CBBs εδI, where parameter ε corresponds to the biomass conversion rate through predation and indicates that there is no total consumption of the adult and immature CBBs that die due to attacks by ants. According to the aforementioned, the population dynamics of the ants were modeled by the following differential equation:(3)$$\frac{dH}{dt}=\left(r+\varepsilon \alpha A+\varepsilon \delta I\right)H\left(1-\frac{H}{k}\right).$$

Thus, the system (4) formed by differential Equations (1)–(3) describes, according to the assumptions proposed, a predator–prey-type interaction between the CBB (*Hypothenemus hampei*) and one of the species of ants that have as part of their diet adult and immature CBBs.
(4)$$\begin{array}{cc} \frac{dA}{dt} & =\omega I-\mu A-\alpha AH\\ \frac{dI}{dt} & =\phi A\left(1-\frac{I}{q}\right)-\omega I-\theta I-\delta IH\\ \frac{dH}{dt} & =\left(r+\varepsilon \alpha A+\varepsilon \delta I\right)H\left(1-\frac{H}{k}\right). \end{array}$$

## 3. Results

### 3.1. Region of Invariance

The study of model (4) started with Proposition 1, where the region of invariance Ω was determined. This is a set of biological interest that is positively invariant with respect to the system’s flow, that is, every solution of the system (4) that starts in Ω remains there throughout t≥0.

**Proposition 1.** *Set* Ω, *given in the following, is a positively invariant region:*
(5)$$\begin{array}{c} \Omega =\left\{\left(A,I,H\right)\in R^{3}:0\leq A\leq \frac{\omega q}{\mu },0\leq I\leq q,0\leq H\leq k\right\}. \end{array}$$

**Proof.** When considering H=0 and then H=k in the third differential equation of the system (4), we obtained
$${\left.\frac{dH}{dt}\right|}_{H=0}={\left.\frac{dH}{dt}\right|}_{H=k}=0.$$This indicated that in those cases there was no variation in *H*; thereby, the population dynamics were determined by variables *A* and *I* on planes H=0 and H=k, respectively. Thus, trajectories with initial conditions on planes H=0 or H=k remained in said planes, as illustrated in Figure 1, where the initial conditions were in the form $\left(A(0),I(0),H(0)\right)=\left(A_{0},I_{0},0\right)$ and $\left(A(0),I(0),H(0)\right)=\left(A_{0},I_{0},k\right)$. In other words, planes H=0 and H=k were invariant.Now, in the first differential equation of the system (4), it was assumed that A=0 and then $A=\frac{\omega q}{\mu }$, which yielded
$${\left.\frac{dA}{dt}\right|}_{A=0}=\omega I\geq 0\;\;\;\;y\;\;\;\;{\left.\frac{dA}{dt}\right|}_{A=\frac{\omega q}{\mu }}=\omega \left(I-q\right)-\frac{q\alpha \omega }{\mu }H\leq 0,$$
respectively, where the last result was guaranteed because I≤q. This meant that trajectories with initial conditions on planes A=0 or $A=\frac{\omega q}{\mu }$ entered region Ω, where the initial conditions were in the form $\left(A(0),I(0),H(0)\right)=\left(0,I_{0},H_{0}\right)$ or $\left(A(0),I(0),H(0)\right)=\left(\frac{\omega q}{\mu },I_{0},H_{0}\right)$, with $I_{0}$ and $H_{0}$ non-negative.Finally, if in the second differential equation of the system (4) I=0 and then I=q were considered, we obtained
$${\left.\frac{dI}{dt}\right|}_{I=0}=\phi A\geq 0\;\;\;\;\;\;y\;\;\;\;\;\;{\left.\frac{dI}{dt}\right|}_{I=q}=-\left(\omega +\theta +\delta H\right)q<0,$$
respectively. This meant that trajectories with initial conditions on planes I=0 or I=q entered region Ω, where the initial conditions were in the form $\left(A(0),I(0),H(0)\right)=\left(A_{0},0,H_{0}\right)$ or $\left(A(0),I(0),H(0)\right)=\left(A_{0},q,H_{0}\right)$, with $A_{0}$ and $H_{0}$ non-negative. □

### 3.2. Equilibrium Points and Stability

An equilibrium point (equilibrium solution) was reached when the rates of change of all system variables were equal to zero. This meant that the variables were neither increasing nor decreasing at that point, resulting in a constant population in equilibrium. To obtain the equilibrium points of the model (4), we set the derivatives equal to zero, and the following was obtained: (6)$$\begin{array}{cc} E_{1} & =\left(0,0,0\right)\\ E_{2} & =\left(\frac{q\omega }{\mu }\left(\frac{B_{0}-1}{B_{0}}\right),q\left(\frac{B_{0}-1}{B_{0}}\right),0\right) \end{array}$$(7)$$\begin{array}{cc} E_{3} & =\left(0,0,k\right)\\ E_{4} & =\left(\frac{q\omega }{k\alpha +\mu }\left(\frac{\Psi_{0}-1}{\Psi_{0}}\right),q\left(\frac{\Psi_{0}-1}{\Psi_{0}}\right),k\right). \end{array}$$


Here, $B_{0}$ and $\Psi_{0}$ are the thresholds on which the biological sense depends with equilibrium points $E_{2}$ and $E_{4}$, respectively, and are given explicitly by
(8)$$\begin{array}{cc} B_{0} & =\frac{\phi \omega }{\mu (\theta +\omega )} \end{array}$$
(9)$$\begin{array}{cc} \Psi_{0} & =\frac{\phi \omega }{\left(k\alpha +\mu )(k\delta +\theta +\omega \right)}. \end{array}$$

Expressions like (8) and (9) are sometimes used in studies of population ecology, for example, in [23], where this type of threshold was denoted *net reproduction rate* and whose definition, adapted to the context of the CBB, corresponded to the number of female CBBs produced by a female CBB in the absence of the predator. Additionally, they have been used in works on the CBB, such as [24,25]. The stability analysis of the equilibrium points was performed based on these thresholds. An equilibrium point is considered local and asymptotically stable if the solutions of the system of differential equations, starting near that point, converge to it as time tends to infinity. This implies that if a population is near a local and asymptotically stable equilibrium point, over time the population will stabilize at that point. In ecological terms, this means that the population will remain at that equilibrium point in the long term. An equilibrium point is considered unstable if the solutions of the system of differential equations, starting near that point, move away from it as time progresses. In other words, any perturbation in the population will cause it to move away from the equilibrium point. In the context of population ecology, this may imply that the population cannot sustain itself at that point and will experience significant changes over time.

Now that the model’s equilibrium points (4) are known explicitly, stability aspects are omitted momentarily to illustrate, in Figure 1, the invariance of the lower and upper boundaries of region Ω given in (5). Figure 1 was elaborated keeping in mind the values of the parameters consigned in Table 1 and considering μ=0.03, showing the trajectories of system (4) with initial conditions on planes H=0 (Figure 1A) and H=k (Figure 1B), surfaces that contained the lower and upper boundaries, respectively, of the invariance region Ω. In the first case, the trajectories tended to the equilibrium point $E_{2}$ on plane H=0 and, in the second case, they tended to the equilibrium point $E_{4}$ on plane H=k.

The following subsection presents and proves Proposition 2, whereby the stability, collision, and biological sense of the equilibrium points $E_{1}$, $E_{2}$, $E_{3}$, and $E_{4}$ were studied using the values assumed by the positive thresholds $B_{0}$ and $\Psi_{0}$ given in (8) and (9), respectively, following the methodology applied in works like [18,26]. The analysis of the equilibrium point $E_{3}$ when $\Psi_{0}=1$ is left out, because it will be discussed independently in Proposition 4.

**Proposition 2.** *The stability of equilibrium points $E_{1}$, $E_{2}$, $E_{3}$, and $E_{4}$ of system* (4) *depends on thresholds $B_{0}$ and $\Psi_{0}$ given in* (8) *and* (9), *respectively, in the following manner:*
*1.* *Equilibrium point $E_{1}$ is unstable.**2.* *Equilibrium point $E_{2}$ has no biological sense if $B_{0}<1$, collides with $E_{1}$ if $B_{0}=1$, and is unstable if $B_{0}>1$.**3.* *Equilibrium point $E_{3}$ is local and asymptotically stable if $\Psi_{0}<1$ and is unstable if $\Psi_{0}>1$.**4.* *Equilibrium point $E_{4}$ has no biological sense if $\Psi_{0}<1$, collides with $E_{3}$ if $\Psi_{0}=1$, and is local and asymptotically stable if $\Psi_{0}>1$.*

**Proof.** The Jacobian matrix of system (4), denoted by J(A,I,H), is given by
(10)$$\left[\begin{array}{ccc} -\mu -\alpha H & \omega & -\alpha A\\ \phi \left(1-\frac{I}{q}\right) & -\frac{\phi A}{q}-\omega -\theta -\delta H & -\delta I\\ \varepsilon \alpha H\left(1-\frac{H}{k}\right) & \varepsilon \delta H\left(1-\frac{H}{k}\right) & \left(r+\varepsilon \alpha A+\varepsilon \delta I\right)\left(1-\frac{2H}{k}\right) \end{array}\right]$$The demonstration consisted in evaluating the Jacobian matrix (10) in each equilibrium point and determining the eigenvalues. For equilibrium points that were hyperbolic, stability conclusions were obtained from the sign of the real parts of the eigenvalues, following a methodology based on the application of the Hartman–Grobman theorem [27]. For nonhyperbolic equilibrium points, the direct method of Liapunov was used. It is important to clarify that the expression $\lambda_{i}^{\left(j\right)}$ with i=1,2,3 and j=1,2,3,4 corresponds to the eigenvalue *i* of the Jacobian matrix (10) evaluated in equilibrium point $E_{j}$.
The Jacobian matrix (10) evaluated in equilibrium point $E_{1}=\left(0,0,0\right)$ has the following eigenvalues:
(11)$$\begin{array}{cc} \lambda_{1}^{\left(1\right)} & =r\\ \lambda_{2}^{\left(1\right)} & =-\frac{1}{2}\left[\left(\theta +\mu +\omega \right)+\sqrt{{\left(\theta -\mu +\omega \right)}^{2}+4\phi \omega }\right] \end{array}$$
(12)$$\begin{array}{cc} \lambda_{3}^{\left(1\right)} & =-\frac{1}{2}\left[\left(\theta +\mu +\omega \right)-\sqrt{{\left(\theta -\mu +\omega \right)}^{2}+4\phi \omega }\right]. \end{array}$$By the form of the radicand, it was concluded that the three eigenvalues were real. All the more, the signs of some of the eigenvalues were evident, particularly $\lambda_{1}^{\left(1\right)}>0$ and $\lambda_{2}^{\left(1\right)}<0$. However, the sign of $\lambda_{3}^{\left(1\right)}$ could not be determined immediately, which is why the radicands were expressed in terms of threshold $B_{0}$ to establish conditions, that is:
$$\begin{array}{cc} \lambda_{2}^{\left(1\right)} & =-\frac{1}{2}\left[\left(\theta +\mu +\omega \right)+\sqrt{{\left(\theta +\mu +\omega \right)}^{2}+4\mu \left(\theta +\omega \right)\left(B_{0}-1\right)}\right]\\ \lambda_{3}^{\left(1\right)} & =-\frac{1}{2}\left[\left(\theta +\mu +\omega \right)-\sqrt{{\left(\theta +\mu +\omega \right)}^{2}+4\mu \left(\theta +\omega \right)\left(B_{0}-1\right)}\right]. \end{array}$$Note that if $B_{0}<1$, then $\lambda_{3}^{\left(1\right)}<0$; if $B_{0}>1$, then $\lambda_{3}^{\left(1\right)}>0$; but, as $\lambda_{1}^{\left(1\right)}>0$ y $\lambda_{2}^{\left(1\right)}<0$, it was concluded that in any case, $E_{1}$ is an unstable hyperbolic equilibrium point of the saddle type. However, for the case $B_{0}=1$, $E_{1}$ is a nonhyperbolic equilibrium point, since $\lambda_{3}^{\left(1\right)}=0$. In effect, the eigenvalues are now:
$$\lambda_{1}^{\left(1\right)}=r,\;\;\;\;\lambda_{2}^{\left(1\right)}=-\left(\theta +\mu +\omega \right),\;\;\;\;\lambda_{3}^{\left(1\right)}=0.$$Bearing in mind the stability results obtained above, it was plausible to think that if $B_{0}=1$, equilibrium point $E_{1}$ would be unstable, but the classical qualitative theory was insufficient to verify this idea due to the loss of hyperbolicity; hence, the instability of equilibrium point $E_{1}$ was shown using the direct method of Liapunov [28], whose conclusion was valid independently of threshold $B_{0}$.Let *U* be a open set given by
$$U=\{\left(A,I,H\right)\in R^{3}:H>0\},$$
for which $E_{1}=$(0,0,0) belongs to the closure of *U*. Let *S* be a neighborhood of $E_{1}$ given by
(13)$$S=\left\{\left(A,I,H\right)\in R^{3}:A^{2}+I^{2}+H^{2}<\frac{\rho^{2}}{4}\right\},$$
where
$$\rho =\min\left\{\frac{r}{\sqrt{\varepsilon^{2}\alpha^{2}+\varepsilon^{2}\delta^{2}}},k\right\},$$
and $\frac{r}{\sqrt{\varepsilon^{2}\alpha^{2}+\varepsilon^{2}\delta^{2}}}$ is the distance from the origin to the line of equation r+εαA+εδI=0, shown in Figure 2. And let *V* be the scalar field
V(A,I,H)=H,
which satisfies(a)     V>0 en $U\cap S\setminus \left\{E_{1}\right\}$;(b)$\;\;\;\;\;\dot{V}\left(A,I,H\right)=\dot{H}=\left(r+\varepsilon \alpha A+\varepsilon \delta I\right)H\left(1-\frac{H}{k}\right)>0$ en $U\cap S\setminus \left\{E_{1}\right\}$;(c)     V(0,0,0)=0;(d)     V=0 on that part of the boundary of *U* inside *S*.Therefore, the equilibrium point $E_{1}$ is unstable, according to the direct method of Liapunov for instability.
Figure 2Projection of the neighborhood *S* (13) to the plane AI when ρ=rε2α2+ε2δ2.
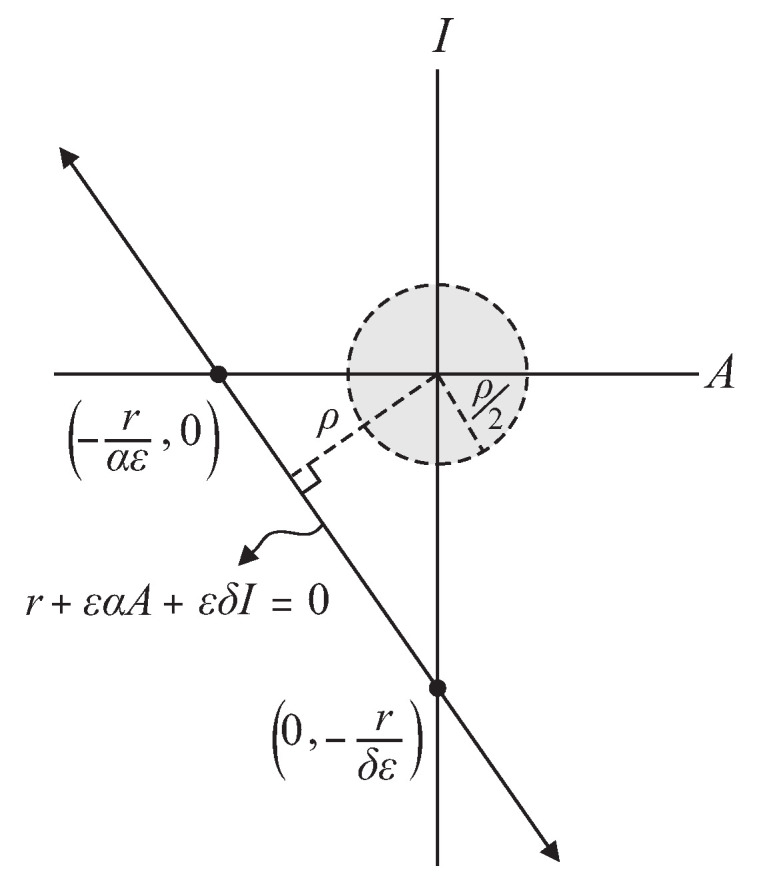

From (6), it is evident that equilibrium point E2 loses its biological sense in the first and second component if B0<1. To analyze the other two cases, the Jacobian matrix (10) was evaluated in equilibrium E2, whose eigenvalues were used to obtain the real value, as observed in the following:
(14)λ1(2)=qεϕω(θ+ω)(αω+δμ)(B0−1)+rλ2(2)=−12μμ2+ϕω+(μ2−ϕω)2+4μ3(θ+ω)=−12μμ2+ϕω+(μ2+ωϕ)2−4μ3(θ+ω)B0−1.
(15)λ3(2)=−12μμ2+ϕω−(μ2−ϕω)2+4μ3(θ+ω)=−12μμ2+ϕω−(μ2+ωϕ)2−4μ3(θ+ω)B0−1.Now, λ2(2)<0 in all cases, and condition B0>1 is sufficient for λ1(2)>0 and implies λ3(2)<0, which allowed us to conclude that E2 is unstable and, specifically, a saddle. In addition, if B0=ϕωμ(θ+ω)=1, we could determine from (6) that E1=E2, i.e., the equilibriums collide. Further, the prior expressions were simplified in the following manner:
λ1(2)=rλ2(2)=−12μμ2+ϕω+μ2+ϕω=−μ2+μθ+ωμ=−θ+μ+ωλ3(2)=−12μμ2+ϕω−μ2+ϕω=0,
which coincides with the results from the previous case that have already been discussed.The Jacobian matrix (10) evaluated in equilibrium E3=(0,0,k) has the following eigenvalues:
(16)λ1(3)=−rλ2(3)=−12Δ+(kα+μ−kδ−θ−ω)2+4ϕω
(17)λ3(3)=−12Δ−(kα+μ−kδ−θ−ω)2+4ϕω
(18)=−12Δ−Δ2+4(kδ+θ+ω)(kα+μ)(Ψ0−1),
where Δ=kα+kδ+θ+μ+ω, and it is clear from (16) and (17) that the eigenvalues are real. It is observed that λ1(3)<0 and λ2(3)<0 in all cases, and in (18) it is noted that the condition Ψ0<1 implies that λ3(3)<0; then, equilibrium E3 is local and asymptotically stable. Furthermore, note from (18) that if Ψ0>1, then λ3(3)>0, and since λ1(3)<0 and λ2(3)<0, it was concluded that equilibrium E3 is unstable and, specifically, a saddle.It is immediately obvious from (7) that equilibrium point E4 loses its biological sense in the first and second component if Ψ0<1. To study the two other cases, the Jacobian matrix (10) was evaluated in equilibrium E4, obtaining, as in previous cases, a J(E4) matrix whose eigenvalues are
(19)λ1(4)=−1ϕωqε(kδ+θ+ω)(αω+kαδ+δμ)(Ψ0−1)+rϕωλ2(4)=−Λ2+ϕω+Λ2−ϕω2+4kδ+θ+ωΛ32Λ
(20)λ3(4)=−Λ2+ϕω−Λ2−ϕω2+4kδ+θ+ωΛ32Λ,
where Λ=kα+μ and, equivalently,
(21)$$\lambda_{3}^{\left(4\right)}=-\frac{\Lambda^{2}+\phi \omega -\sqrt{{\left(\Lambda^{2}+\phi \omega \right)}^{2}-4\left(k\delta +\theta +\omega \right)\Lambda^{3}\left(\Psi_{0}-1\right)}}{2\Lambda }.$$From (19) and (20), it was deducted that the eigenvalues are real. Note in (7) that if $\Psi_{0}=1$, then $E_{4}=E_{3}$, that is, $E_{4}$ collides with $E_{3}$. Finally, if $\Psi_{0}>1$ in (21), it is easy to see that $\lambda_{3}^{\left(4\right)}<0$, but since $\lambda_{1}^{\left(4\right)}<0$ and $\lambda_{2}^{\left(4\right)}<0$, it was concluded that equilibrium point $E_{4}$ is local and asymptotically stable.□

**Remark 1.** *In the demonstration of the four cases contemplated in Proposition 2, it was verified that all the eigenvalues are real; specifically, in expressions* (11)*,* (12)*,* (14)–(17)*,* (19)*, and* (20)*, it was evidenced that no possibility exists of having complex eigenvalues, because all the radicands are positive.*

Proposition 3 analyzed the stability results obtained, but now having as reference parameter μ, the death rate of adult CBBs due to factors other than predation. As such, the stability conditions of Proposition 2, expressed in terms of thresholds $B_{0}$ and $\Psi_{0}$, were rewritten isolating parameter μ, which, as observed in Table 1, had no fixed value, given that it corresponded to the bifurcation (branch) parameter. Note that μ is present in both thresholds and is important from the agricultural-ecological point of view, given that its variation is supported in the modification of its value by the intervention of the coffee grower in the plantation implementing cultural and/or chemical control. It is important to keep in mind in Proposition 3 the inequality $\Psi_{0}<B_{0}$; in effect, from (8) and (9), we obtained
(22)$$\Psi_{0}=\frac{\phi \omega }{k\alpha (k\delta +\theta +\omega )+k\delta \mu +\mu (\theta +\omega )}<\frac{\phi \omega }{\mu (\omega +\theta )}=B_{0}.$$

Furthermore,
$$B_{0}=1\Leftrightarrow \mu =\mu^{**}=\frac{\phi \omega }{\theta +\omega }$$
(23)$$\Psi_{0}=1\Leftrightarrow \mu =\mu^{*}=\frac{\phi \omega }{k\delta +\theta +\omega }-k\alpha ,$$
and it is clear that
$$\mu^{*}=\frac{\phi \omega }{k\delta +\theta +\omega }-k\alpha <\frac{\phi \omega }{\theta +\omega }=\mu^{**}.$$

From the perspective of population ecology, Proposition 3 shows the influence of the value of the parameter μ, the death rate of adult CBBs due to factors other than predation, on population dynamics. In the first case, when $\mu <\mu^{*}$, the equilibrium point of coexistence $E_{4}$ is local and asymptotically stable. This means that, over time, the populations of adult CBBs, immature CBBs, and ants tend to move towards the respective components of the equilibrium point $E_{4}$ given in (7), the only one of the four equilibrium points whose components are all different from zero; that is, $\mu <\mu^{*}$ is the condition that had to be satisfied so that the three populations can coexist. Then, by increasing the value of the parameter μ, the population dynamics are modified by the collision of the equilibrium points or by the loss of its biological sense; however, note that the persistence equilibrium point $E_{3}=\left(0,0,k\right)$ remains locally and asymptotically stable, which means that with time, the populations of adult CBBs, immature CBBs, and ants tend towards the respective components of the equilibrium point $E_{3}$, leading to the extinction of the CBB population and the persistence of the ant population, at least under the considerations of this model. This situation translates into a good outlook for coffee growers, because the condition $\mu \geq \mu^{*}$ (the case $\mu =\mu^{*}$ was studied in Proposition 4) is relatively easy to guarantee, since the value of $\mu^{*}$ is low due to the presence of the carrying capacity of the predatory ant *k* in the expression $\mu^{*}=\frac{\phi \omega }{k\delta +\theta +\omega }-k\alpha$. However, this very favorable situation for coffee growers is due to predation by ants, which is a fact discussed later. All this discussion is complemented in the corresponding section on numerical simulations.

**Proposition 3.** *The stability of equilibrium points $E_{1}$, $E_{2}$, $E_{3}$, and $E_{4}$ of the system* (4) *depends on parameter μ, the death rate of adult CBBs due to factors other than predation, in the following manner:*
*1.* *If $\mu <\mu^{*}$ or, equivalently, $1<\Psi_{0}<B_{0}$, then equilibrium points $E_{1}$, $E_{2}$, and $E_{3}$ are unstable, and equilibrium point $E_{4}$ is local and asymptotically stable.**2.* *If $\mu =\mu^{*}$ or, equivalently, $1=\Psi_{0}<B_{0}$, then equilibrium points $E_{1}$ and $E_{2}$ are unstable, and equilibrium point $E_{4}$ collides with $E_{3}$.**3.* *If $\mu^{*}<\mu <\mu^{**}$ or, equivalently, $\Psi_{0}<1<B_{0}$, then equilibrium points $E_{1}$ and $E_{2}$ are unstable, equilibrium point $E_{3}$ is local and asymptotically stable, and equilibrium point $E_{4}$ lacks biological sense.**4.* *If $\mu =\mu^{**}$ or, equivalently, $\Psi_{0}<B_{0}=1$, then equilibrium point $E_{1}$ is unstable, equilibrium point $E_{2}$ collides with $E_{1}$, equilibrium point $E_{3}$ is local and asymptotically stable, and equilibrium point $E_{4}$ lacks biological sense.**5.* *If $\mu >\mu^{**}$ or, equivalently, $\Psi_{0}<B_{0}<1$, then equilibrium point $E_{1}$ is unstable, equilibrium point $E_{2}$ lacks biological sense, equilibrium point $E_{3}$ is local and asymptotically stable, and equilibrium point $E_{4}$ lacks biological sense.*

Note that in Propositions 2 and 3, no conclusions were obtained for the stability of the equilibrium point $E_{3}$ when (23) is satisfied. This is because a bifurcation occurs in equilibrium point $E_{3}$, and $\mu =\mu^{*}$ is the bifurcation parameter, as shown in the following proposition:

**Proposition 4.** *System* (4) *experiences a transcritical bifurcation in equilibrium point $E_{3}$ when parameter μ goes through the bifurcation value $\mu =\mu^{*}=\frac{\phi \omega }{k\delta +\theta +\omega }-k\alpha$.*

**Proof.** The demonstration of this consisted in verifying that the conditions of the Sotomayor theorem [27] are satisfied, specifically for case (ii). In the first place, we observed from (18) that when (23) is satisfied, equilibrium point $E_{3}$ is nonhyperbolic given that $\lambda_{3}^{\left(3\right)}=0$. Moreover, the Jacobian matrix $J(E_{3})$ has the following eigenvalues:
$$\lambda_{1}^{\left(3\right)}=-r,\;\;\;\;\lambda_{2}^{\left(3\right)}=-\left(k\alpha +k\delta +\theta +\mu +\omega \right),\;\;\;\;\lambda_{3}^{\left(3\right)}=0.$$In this case, $v={\left(\left(k\delta +\theta +\omega \right)/\phi ,1,0\right)}^{T}$ and $w={\left(\left(k\delta +\theta +\omega \right)/\omega ,1,0\right)}^{T}$ are the eigenvectors associated with the null eigenvalue of matrices $J(E_{3})$ and $J^{T}\left(E_{3}\right)$, respectively. Additionally, $f_{\mu }\left(A,I,H,\mu \right)={\left(-A,0,0\right)}^{T}$, where f is the system’s (4) vector field; thus,
(24)$$w^{T}f_{\mu }\left(0,0,k,\mu^{*}\right)=\left(\frac{k\delta +\theta +\omega }{\omega },1,0\right)\left(\begin{array}{c} 0\\ 0\\ 0 \end{array}\right)=0.$$Secondly, we determined that
(25)$$\begin{array}{cc} w^{T}\left[Df_{\mu }\left(0,0,k,\mu^{*}\right)v\right] & =\left(\frac{k\delta +\theta +\omega }{\omega },1,0\right)\left[\left(\begin{array}{ccc} -1 & 0 & 0\\ 0 & 0 & 0\\ 0 & 0 & 0 \end{array}\right)\left(\begin{array}{c} \frac{k\delta +\theta +\omega }{\phi }\\ 1\\ 0 \end{array}\right)\right]\\ \; & =-\frac{1}{\phi \omega }{\left(k\delta +\theta +\omega \right)}^{2}\neq 0. \end{array}$$
Finally, the methodology used in [27] was used to obtain the following result:
$$D^{2}f\left(A,I,H,\mu \right)\left(v,v\right)=\left(\begin{array}{c} 0\\ -\frac{2\left(k\delta +\theta +\omega \right)}{q}\\ 0 \end{array}\right),$$
where $D^{2}f\left(x\right)\left(v,v\right)=D\left(Df\left(x\right)v\right)v$. Then,
(26)$$\begin{array}{cc} w^{T}\left[D^{2}f\left(0,0,k,\mu^{*}\right)\left(v,v\right)\right] & =\left(\frac{k\delta +\theta +\omega }{\omega },1,0\right)\left(\begin{array}{c} 0\\ -\frac{2\left(k\delta +\theta +\omega \right)}{q}\\ 0 \end{array}\right)\\ \; & =-\frac{2\left(k\delta +\theta +\omega \right)}{q}\neq 0. \end{array}$$Upon verifying conditions (24)–(26), the proposition was demonstrated according to Sotomayor’s theorem [27]. □

To summarize, Figure 3 condenses the stability results obtained in Propositions 3 and 4. It is important to clarify that equilibrium point $E_{4}$ does not disappear after colliding with equilibrium point $E_{3}$ when $\mu =\mu^{*}$; in reality, it is unstable but lacks biological sense according to Proposition 3; hence, this situation was not considered in Figure 3.

This section ends with a result that allowed us to achieve the full stability analysis of system (4). Note in Figure 1A that on plane H=0, equilibrium $E_{1}$ is unstable and equilibrium $E_{2}$ is local and asymptotically stable. However, under certain conditions, this situation can be modified; in fact, bifurcation occurs, as shown in Proposition (5).

In Proposition 5, system (4) on plane H=0 is analyzed, giving rise to system (27) and translating into a population dynamic with no ant population. On the other hand, the trivial equilibrium point ${\bar{E}}_{1}=\left(0,0\right)$ is the only equilibrium point of system (27) if $\mu =\mu^{**}=\frac{\phi \omega }{\theta +\omega }$. It is the only point of the equilibrium of system (27) with a biological sense if $\mu >\mu^{**}$; in addition, it is local and asymptotically stable. In other words, in the absence of a predatory ant, the condition $\mu \geq \mu^{**}=\frac{\phi \omega }{\theta +\omega }$ must be satisfied (i.e., the parameter μ must reach a value much higher than that exhibited in Proposition 4) in order to eradicate the CBB, under the considerations of this model. For coffee farmers, this means having to guarantee that control methods other than biological approaches significantly increase the death rate of adult CBBs due to factors other than predation.

**Proposition 5.** *On plane H=0, system* (4) *experiences a transcritical bifurcation in equilibrium point $E_{1}$ when parameter μ goes through the bifurcation value $\mu =\mu^{**}=\frac{\phi \omega }{\theta +\omega }$.*

**Proof.** The dynamic of system (4) on plane H=0 is described by the following reduced system:
(27)$$\begin{array}{cc} \frac{dA}{dt} & =\omega I-\mu A\\ \frac{dI}{dt} & =\phi A\left(1-\frac{I}{q}\right)-\omega I-\theta I, \end{array}$$
whose equilibrium points are
$${\bar{E}}_{1}=\left(0,0\right)\;\;\;\;{\bar{E}}_{2}=\left(\frac{q\omega }{\mu }\left(\frac{B_{0}-1}{B_{0}}\right),q\left(\frac{B_{0}-1}{B_{0}}\right)\right)$$
where $B_{0}$ is given by (8). It is evident that equilibrium point ${\bar{E}}_{2}$ loses its biological sense if $B_{0}<1$, but it is biologically admissible if $B_{0}\geq 1$; specifically, equilibrium ${\bar{E}}_{2}$ collides with ${\bar{E}}_{1}$ if $B_{0}=1$.Proceeding with the classical stability analysis, the Jacobian matrix associated with system (27) was evaluated in equilibrium point ${\bar{E}}_{1}$, that is, $J({\bar{E}}_{1})$, and the following eigenvalues were obtained:
$$\begin{array}{cc} {\bar{\lambda }}_{1}^{\left(1\right)} & =-\frac{1}{2}\left[\left(\theta +\mu +\omega \right)+\sqrt{{\left(\theta -\mu +\omega \right)}^{2}+4\phi \omega }\right]\\ \; & =-\frac{1}{2}\left[\left(\theta +\mu +\omega \right)+\sqrt{{\left(\theta +\mu +\omega \right)}^{2}+4\mu \left(\theta +\omega \right)\left(B_{0}-1\right)}\right]\\ {\bar{\lambda }}_{2}^{\left(1\right)} & =-\frac{1}{2}\left[\left(\theta +\mu +\omega \right)-\sqrt{{\left(\theta -\mu +\omega \right)}^{2}+4\phi \omega }\right]\\ \; & =-\frac{1}{2}\left[\left(\theta +\mu +\omega \right)-\sqrt{{\left(\theta +\mu +\omega \right)}^{2}+4\mu \left(\theta +\omega \right)\left(B_{0}-1\right)}\right] \end{array}$$Conducting the same process with equilibrium point ${\bar{E}}_{2}$, the eigenvalues of $J({\bar{E}}_{2})$ were obtained, which are given by:
$$\begin{array}{cc} {\bar{\lambda }}_{1}^{\left(2\right)} & =-\frac{1}{2\mu }\left[\mu^{2}+\phi \omega +\sqrt{{\left(\mu^{2}-\phi \omega \right)}^{2}+4\mu^{3}\left(\theta +\omega \right)}\right]\\ & =-\frac{1}{2\mu }\left[\mu^{2}+\phi \omega +\sqrt{{\left(\mu^{2}+\omega \phi \right)}^{2}-4\mu^{3}\left(\theta +\omega \right)\left(B_{0}-1\right)}\right]\\ {\bar{\lambda }}_{2}^{\left(2\right)} & =-\frac{1}{2\mu }\left[\mu^{2}+\phi \omega -\sqrt{{\left(\mu^{2}-\phi \omega \right)}^{2}+4\mu^{3}\left(\theta +\omega \right)}\right]\\ & =-\frac{1}{2\mu }\left[\mu^{2}+\phi \omega -\sqrt{{\left(\mu^{2}+\omega \phi \right)}^{2}-4\mu^{3}\left(\theta +\omega \right)\left(B_{0}-1\right)}\right]. \end{array}$$From the previous results, it was concluded that if $B_{0}\neq 1$, equilibrium points ${\bar{E}}_{1}$ and ${\bar{E}}_{2}$ are hyperbolic. Furthermore:
If $B_{0}=\frac{\phi \omega }{\mu \left(\theta +\omega \right)}>1\;\Leftrightarrow \mu <\mu^{**}=\frac{\phi \omega }{\theta +\omega }$−$\;\;\;\;{\bar{\lambda }}_{1}^{\left(1\right)}<0$ and ${\bar{\lambda }}_{2}^{\left(1\right)}>0$; hence, equilibrium point ${\bar{E}}_{1}$ is unstable.−$\;\;\;\;{\bar{\lambda }}_{1}^{\left(2\right)}<0$ and ${\bar{\lambda }}_{2}^{\left(2\right)}<0$; hence, equilibrium point ${\bar{E}}_{2}$ is local and asymptotically stable.If $B_{0}=\frac{\phi \omega }{\mu \left(\theta +\omega \right)}<1\Leftrightarrow \mu >\mu^{**}=\frac{\phi \omega }{\theta +\omega }$−$\;\;\;\;{\bar{\lambda }}_{1}^{\left(1\right)}<0$ and ${\bar{\lambda }}_{2}^{\left(1\right)}<0$; hence, equilibrium point ${\bar{E}}_{1}$ is local and asymptotically stable.−$\;\;\;\;{\bar{\lambda }}_{1}^{\left(2\right)}<0$ and ${\bar{\lambda }}_{2}^{\left(2\right)}>0$; hence, equilibrium point ${\bar{E}}_{2}$ is unstable, although without biological sense.Thus, if $B_{0}=\frac{\phi \omega }{\mu \left(\theta +\omega \right)}=1\Leftrightarrow \mu =\mu^{**}=\frac{\phi \omega }{\theta +\omega }$, equilibrium point ${\bar{E}}_{2}$ collides with ${\bar{E}}_{1}$, and this equilibrium point is no longer hyperbolic, because the eigenvalues of $J({\bar{E}}_{1})$ are
$${\bar{\lambda }}_{1}^{\left(1\right)}=-\left(\theta +\mu^{**}+\omega \right)=-\frac{{\left(\theta +\omega \right)}^{2}+\phi \omega }{\theta +\omega }\;\;\;\;\;\;\;{\bar{\lambda }}_{2}^{\left(1\right)}=0.$$To show the occurrence of transcritical bifurcation, we proceeded as in the demonstration of Proposition 4, that is, by verifying the fulfillment of the conditions of the Sotomayor’s theorem [27], specifically for case (ii). In this case, $v={\left(\left(\theta +\omega \right)/\phi ,1\right)}^{T}$ and $w={\left(\left(\theta +\omega \right)/\omega ,1\right)}^{T}$ are the eigenvectors associated with the null eigenvalue of matrices $J({\bar{E}}_{1})$ and $J^{T}\left({\bar{E}}_{1}\right)$, respectively. Moreover, $f_{\mu }\left(A,I,\mu \right)={\left(-A,0\right)}^{T}$, where f is the vectorial field associated with system (27). In this way, the necessary inputs are available to establish the following results:
(28)$$w^{T}f_{\mu }\left(0,0,\mu^{**}\right)=\left(\frac{\theta +\omega }{\omega },1\right)\left(\begin{array}{c} 0\\ 0 \end{array}\right)=0.$$
(29)$$\begin{array}{cc} w^{T}\left[Df_{\mu }\left(0,0,\mu^{**}\right)v\right] & =\left(\frac{\theta +\omega }{\omega },1\right)\left[\left(\begin{array}{cc} -1 & 0\\ 0 & 0 \end{array}\right)\left(\begin{array}{c} \frac{\theta +\omega }{\phi }\\ 1 \end{array}\right)\right]\\ \; & =-\frac{1}{\phi \omega }{\left(\theta +\omega \right)}^{2}\neq 0. \end{array}$$
(30)$$\begin{array}{cc} w^{T}\left[D^{2}f\left(0,0,\mu^{**}\right)\left(v,v\right)\right] & =\left(\frac{\theta +\omega }{\omega },1\right)\left(\begin{array}{c} 0\\ -\frac{2\left(\theta +\omega \right)}{q} \end{array}\right)\\ \; & =-\frac{2\left(\theta +\omega \right)}{q}\neq 0. \end{array}$$Once conditions (28)–(30) were verified, the proposition was demonstrated. □

### 3.3. Numerical Simulations

Seeking to illustrate the stability results obtained in Propositions 3 and 4, numerical simulations of model (4) were carried out using the values of the parameters shown in Table 1, which were chosen only for illustration purposes and are not necessarily realistic [29]. In fact, although some of them were taken from the literature, they cannot be assumed to be standard values, given that other research could offer different figures; in this regard, ref. [5] indicated that considerable differences exist with respect to information on the duration of the states that constitute the life cycle of the CBB, which is due principally to the variation in the environmental conditions of diverse studies, most importantly temperature.

**Table 1 insects-14-00675-t001:** Values of parameters used in the simulations of model (4).

Parameter	Description	Value	Ref.
*q*	Carrying capacity of the immature CBBs	700	
*k*	Carrying capacity of the predatory ants	q/20	
*r*	Intrinsic growth rate of predatory ants	0.01	
α	Predation rate of ants on adult CBBs	0.01	[30]
δ	Predation rate of ants on immature CBBs	0.01	[30]
ε	Biomass conversion rate through predation	0.2	
θ	Natural death rate of immature CBBs	0.23	[31]
μ	Death rate of adult CBBs due to factors other than predation	Variable	
ϕ	Oviposition rate of adult CBB females	2	[5]
ω	Development rate from immature stage to adult stage	0.17	[31]

Prior to conducting the simulations, it was necessary to discuss parameter *q*, corresponding to the carrying capacity of the immature CBBs. Regarding this parameter, some authors hold that its real value is unavailable. For example, the authors of [5] stated that it is impossible conduct CBB censuses, given the amount of beans existing on a tree, the high density of trees per hectare, and the economic unfeasibility; furthermore, the authors of [4] stated that the number of CBBs per hectare can reach millions. Bearing in mind the above, in Table 1 the value for *q*, which was hypothetically assigned, is quite small in relation to what could be the real figure but adequate for the aim of this section. It is important to mention that for the carrying capacity of the predatory ants, a figure proportional to parameter *q* was considered, specifically, a twentieth part. This was based on the premise that in the area cultivated with coffee, the number of predatory ants would be lower than that of CBBs, which was plausible considering that in the assumptions established, the interactions were assumed to occur with a single species of predatory ant.

Complementarily, but with respect to adult CBBs, since a carrying capacity figure was lacking, the upper bound of *A* was assumed to be in the invariance region (5) as the maximum number of adult CBBs considered, that is, ωq/μ. However, this expression contained μ, the parameter that, according to Proposition 3, would be modified; hence, μ=0.03 was assumed, the lowest of the values considered in all the simulations. Thus, ωq/μ=3967.66, approximately.

The simulations started with Figure 4A, which shows the four equilibrium points. Of these, only $E_{4}$ was local and asymptotically stable, which occurred for $0.03=\mu <\mu^{*}=0.1033$, that is, the three populations considered in the model (adult CBBs, immature CBBs, and ants from a predatory species) coexisted, while no more than approximately 409.76 (3967.66×0.1033) adult CBBs died in one day due to factors other than predation. The three initial conditions (10,20,1), (3800,700,1), and (5,1,32) were set close to equilibrium points $E_{1}$, $E_{2}$, and $E_{3}$, respectively, to exhibit the instability of the latter, which was evidenced from the direction of the trajectories.

Figure 4B shows only three equilibriums, because $E_{4}$ collided with $E_{3}$ and transferred its stability, which occurred precisely when $\mu =\mu^{*}=0.1033$, that is, if approximately 409.86 adult CBBs died daily due to factors other than predation. In this situation, the three populations in question stopped coexisting, because the local and asymptotic stability of $E_{3}=\left(0,0,k\right)$ indicated that both CBB populations were extinguished, and the predatory ant population tended to its carrying capacity.

**Figure 4 insects-14-00675-f004:**
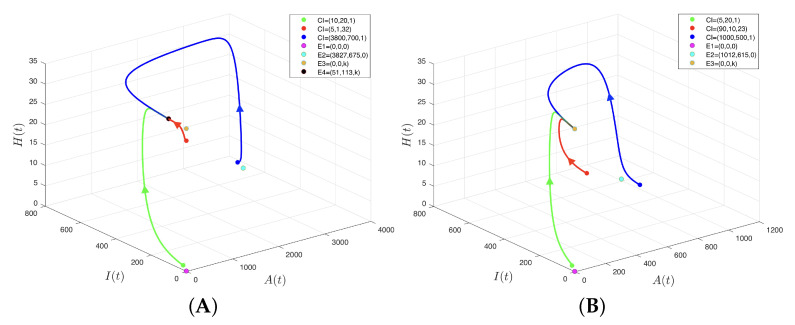
Phase diagram of model (4). (**A**) Case $\mu <\mu^{*}=\frac{\phi \omega }{k\delta +\theta +\omega }-k\alpha$ with μ=0.03, used to obtain $\Psi_{0}=1.1930$ and $B_{0}=28.33$; thus, equilibrium points $E_{1}$, $E_{2}$, and $E_{3}$ are unstable, and equilibrium point $E_{4}$ is local and asymptotically stable. (**B**) Case $\mu =\mu^{*}=\frac{\phi \omega }{k\delta +\theta +\omega }-k\alpha$ with μ=0.1033, used to obtain $\Psi_{0}=1$ and $B_{0}=8.2258$; thus, equilibrium points $E_{1}$ and $E_{2}$ are unstable, while equilibrium point $E_{4}$ collides with $E_{3}$, which is local and asymptotically stable. The use of colors aims to differentiate between the equilibrium points and the initial conditions from which the trajectories originate.

Figure 5A shows a similar picture to that of Figure 4. In fact, according to Proposition 3, the coexistence between the species considered in model (4) stops occurring when $\mu \geq \mu^{*}=0.1033$, which is very good for coffee growers. In other words, for both the CBB populations to be extinguished and the predatory ant population to tend to its carrying capacity, the coffee grower must guarantee that 409.86 or more adult CBBs die per day due to factors other than predation, such as the use of alcohol-based handmade traps, which is a form of cultural control, or the moderate use of insecticides.

Figure 5B shows only equilibrium points $E_{1}$ and $E_{3}$, because equilibrium $E_{2}$ collided with equilibrium $E_{1}$ when $\mu =\mu^{**}=0.85$; in fact, it lost its biological sense when $\mu >\mu^{**}=0.85$, but this case was not illustrated graphically. In any case, it was concluded that if a coffee grower guaranteed that 3372.511(3967.66×0.85) or more adult CBBs died per day due to factors other than predation, the extinction of the CBB populations would be achieved, but without the need for intervention by predatory ants. In effect, it is possible that $\mu \geq \mu^{**}=\frac{\phi \omega }{\theta +\omega }$, although α=δ=0. This is a situation that must be avoided in practice, because one of the ways of achieving said levels of μ is to resort to increasing the use of insecticides.

**Figure 5 insects-14-00675-f005:**
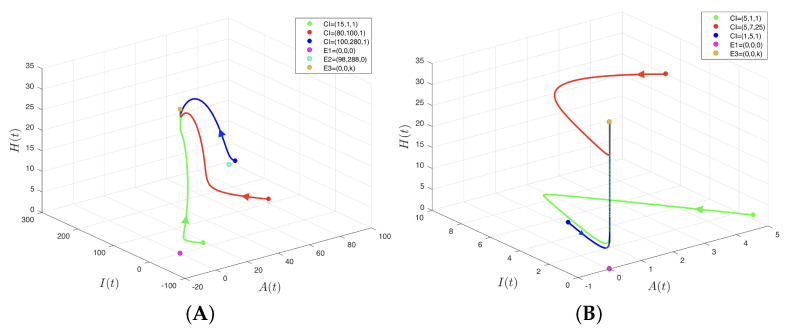
Phase diagram of model (4). (**A**) Case $\frac{\phi \omega }{k\delta +\theta +\omega }-k\alpha =\mu^{*}<\mu <\mu^{**}=\frac{\phi \omega }{\theta +\omega }$ with μ=0.5, used to obtain $\Psi_{0}=0.5333$ and $B_{0}=1.7$; thus, equilibrium points $E_{1}$ and $E_{2}$ are unstable, $E_{3}$ is local and asymptotically stable, and $E_{4}$ loses its biological sense. (**B**) Case $\mu =\mu^{**}=\frac{\phi \omega }{\theta +\omega }$ with μ=0.85, used to obtain $\Psi_{0}=0.3778$ and $B_{0}=1$. Thus, equilibrium point $E_{2}$ collides with $E_{1}$, which is unstable; $E_{3}$ is local and asymptotically stable; and $E_{4}$ lacks biological sense. The use of colors aims to differentiate between the equilibrium points and the initial conditions from which the trajectories originate.

## 4. Discussion

In population ecology, mathematical models are essential due to their importance for elaborating theoretical arguments related to the factors that affect population size [32]. Transcendent evidence was provided by the predator–prey model proposed and studied in this work, given that conditions could be formulated in terms of ecological thresholds and the parameters that determined the coexistence of the three populations considered (adult CBBs, immature CBBs, and ants from a predatory species) or the hypothetical extermination of the pest. It is essential to highlight that in this model, no possibility existed of extinguishing the population of ants, because a realistic assumption was taken into account, and the ants—despite their predator function—were not considered to feed exclusively on CBBs, as occurs in nature. In addition, only natural death was assumed as a growth-inhibiting factor for this population, which was implicit in the intrinsic growth rate, that is, parameter *r*. Note that by omitting predation by ants in the model, that is, making α=δ=0 in system (4), a decoupled model was obtained, wherein the third differential equation, corresponding to the predatory ants, was the classical logistic equation, whose solution with r>0 satisfied H(t)→k. This meant that in the long term, the predatory ants tended towards their carrying capacity.

An important fact to highlight in the decoupled model is that the system formed by the first two differential equations was already analyzed in Proposition 5, specifically, model (27). In this case, the dynamics of system (4) on plane H=0, that is, the interaction of the CBB populations, was analyzed in the absence of a predator. In either case, without predation (α=δ=0) or in the absence of a predator (H=0), the pest was extinguished if the following condition was satisfied:(31)$$B_{0}=\frac{\phi \omega }{\mu \left(\theta +\omega \right)}<1\Leftrightarrow \mu >\mu^{**}=\frac{\phi \omega }{\theta +\omega },$$
because the origin would be a local and asymptotically stable equilibrium point. In other words, the death rate of adult CBBs due to factors other than predation, denoted by μ, had to be very high. According to the values in Table 1, the expression $\mu >\mu^{**}=\frac{\phi \omega }{\theta +\omega }=0.85$ meant that it was necessary to ensure the daily death, without predation, of approximately 3373 adult CBBs, from the total of 3968 individuals considered as the maximum population of adult CBBs, a figure based on the upper bound for state variable *A* in the region of invariance. Such a high death rate for adult CBBs is synonymous with ideal conditions for coffee growers. However, for this to be achieved, it is necessary to implement other practices to regulate CBBs, like chemical control through the application of insecticides, which only affect adult CBBs that are outside the bean. Nevertheless, it is known in the coffee world that this control practice is not recommended due to the collateral damage caused to the plantation’s flora and fauna [5]. This problem, although derived from the analysis of a theoretical model, evidences that to counter the effects of the CBB, it is more efficient to combine different types of controls, a fact that will be emphasized from now on.

From Proposition 3, it was determined that if condition (31) was fulfilled, equilibrium point $E_{1}=\left(0,0,0\right)$ would be unstable, and equilibrium point $E_{3}=\left(0,0,k\right)$ would be local and asymptotically stable, that is, the initial conditions close to $E_{1}$ would originate trajectories that asymptotically approached $E_{3}$. In fact, if condition (31) was fulfilled, any trajectory with initial condition $\left(A_{0},I_{0},H_{0}\right)$ in the invariance region and $H_{0}\neq 0$ would tend asymptotically to equilibrium $E_{3}$. Ecologically, this means that under condition (31), interpreted according to the figures in the previous paragraph, no possibility would exist for the CBB populations to prosper; however, favorable results could also be obtained with respect to a decrease in CBBs under weaker conditions for parameter μ, due to the participation of the ants in the dynamics, fulfilling a predatory role. In effect, Propositions 2 and 3 used a condition in which the predation rates of ants affected adult CBBs and immature CBBs, that is, α and δ, respectively. This condition is given by
(32)$$\Psi_{0}=\frac{\phi \omega }{\left(k\alpha +\mu )(k\delta +\theta +\omega \right)}\leq 1\Leftrightarrow \mu \geq \mu^{*}=\frac{\phi \omega }{k\delta +\theta +\omega }-k\alpha$$
which was sufficient for CBBs and predatory ant populations to not coexist, because the coexistence equilibrium point $E_{4}$ collided with equilibrium $E_{3}$ or lost its biological sense; furthermore, from the other three equilibrium points, only $E_{3}=\left(0,0,k\right)$ was local and asymptotically stable, and, as stated previously, this meant that any trajectory with initial condition $\left(A_{0},I_{0},H_{0}\right)$ in the invariance region with $H_{0}\neq 0$ tended asymptotically to equilibrium $E_{3}$, as observed in Figure 4 and Figure 5.

Using the values from Table 1 in condition (32), we obtained $\mu \geq \mu^{*}=0.1033$, meaning that at least 410(3967.66×0.1033) daily adult CBB deaths due to factors other than predation were required for the pest to be eradicated. This figure is easier to reach for coffee growers compared with the 3373 adult CBBs they would need to eliminate to achieve the same results, i.e., the extinction of the entire CBB population, when not considering the ants’ predatory action in the dynamic. Note from Table 1 that the predation rates of ants against adult CBBs and immature CBBs were α=0.01 and δ=0.01, respectively, but increasing either of these rates reduced the number of adult CBBs that had to be eliminated to achieve the eradication target; in this regard, Table 2 shows different scenarios. Thereby, it is clear that favoring predation rates lessened the effort that had to be put into implementing other types of controls to reach the number of deaths of adult CBBs required to exterminate the pest, at least theoretically. This favoring of the predation of CBBs could be achieved by maintaining ant habitats, as suggested in [14], and/or using attractive solutions, as reported in [33]. It is important to clarify that in a realistic mathematical model, the death rate of adult CBBs due to factors other than predation, represented in this case by parameter μ, cannot be zero, because adult CBBs also die of natural causes.

According to the above, realism was observed in the results obtained in this work and their importance to coffee growing. Furthermore, we showed that, concerning controls against the CBB, IBM is the best option, corresponding in this case to the combination of biological control (represented by ants from species that include in their diets the consumption of CBBs) and other types of control (chemical or cultural, exemplified in plantations through the application of insecticides or handmade traps with alcoholic attractants, respectively). It is essential to mention that in order to simplify the model, we assumed that the CBBs were being attacked by only one of their natural enemies (specifically, ants, acting as predators within the dynamics). However, this was an unrealistic consideration, because fungi, ants, microhymenoptera, birds, nematodes, coleoptera, reptiles, and thrips can all be biological control agents [34]. Furthermore, other predators of CBBs continue to be discovered, such as a green lacewing species, *Chrysoperla externa* (Hagen) [35]. Therefore, as future work, the proposed model could consider incorporating biological control involving the simultaneous action of different species.

## Figures and Tables

**Figure 1 insects-14-00675-f001:**
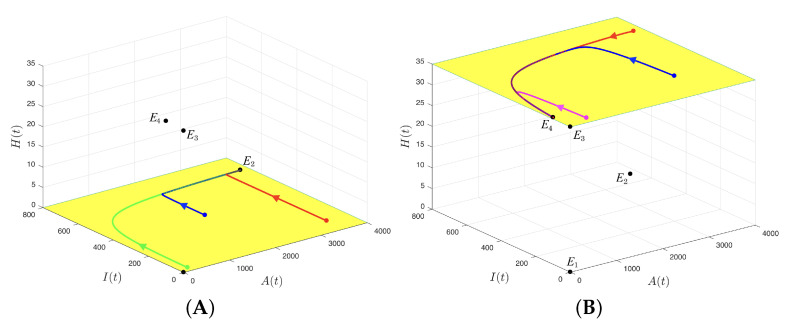
Trajectories of system (4). In (**A**), the trajectories and initial conditions were found on plane H=0, and in (**B**), the trajectories and initial conditions were found on plane H=k. The use of colors aims to differentiate the trajectories on the relevant planes, which were colored in yellow.

**Figure 3 insects-14-00675-f003:**
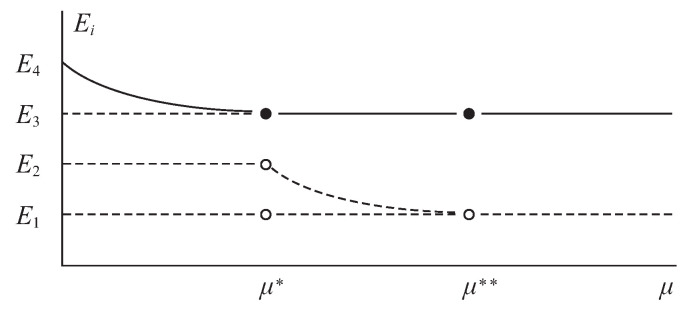
Bifurcation diagram summarizing the stability results of Propositions 3 and 4. The conti-nuous line and the solid circle indicate that the equilibrium point is local and asymptotically stable, while the broken line and hollow circle indicate that the equilibrium point is unstable. In this case, $\mu^{*}$ and $\mu^{**}$ are bifurcation values.

**Table 2 insects-14-00675-t002:** Relationship between predation rates and the death rate of adult CBBs due to other factors.

α	δ	$\mu^{*}$	Number of Adult CBB Deaths Due to Factors Other than Predation
0.01	0.01	0.1033	410
0.011	0.01	0.0831	330
0.01	0.011	0.0683	271
0.011	0.011	0.0481	191

## Data Availability

No new data were created or analyzed in this study. Data sharing is not applicable to this article.

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
