# Peer review of "Mathematical Model for the Biological Control of the Coffee Berry Borer Hypothenemus hampei through Ant Predation"

_insects, 2023, doi:10.3390/insects14080675_

Round 1

Reviewer 1 Report

Review for Insects 2375699

Mathematical Model for the Biological Control of the Coffee Berry Borer Hypothenemus hampei through Ant Predation

General Comments:

This study aims to examine how predation by ants contribute to population regulation of the coffee berry borer, and to what extent other control methods will be needed in order to keep the population at a low level.

My first comment is that the specific aims and questions of the study are not very clearly articulated in the introduction, and therefore, the different propositions (which I assume to be model interations?) are also not very clear. I would suggest that the authors very clearly outline what are the study aims, and what are the different models / parameter values investigated very early on in the manuscript.

Second, I had a lot of questions about the model parameters. I am not a theoretical ecologist, and so I was not able to understand all of the mathematical modeling outlined in the paper. That said, I think that there should be more explanation – in biological terms instead of mathematical terms – to show where the parameter values come from, and what different combinations were tested in the different propositions.

Third, I am not sure if including ants in the model just as one species is realistic, given that we do know that several different species may prey on the CBB. It might be interesting to have a section in the discussion to further explore what may be missing from the model that might also account for increased borer mortality (e.g., predation by birds, attack by entomopathogenic fungi, parasitism by wasps, etc.). The conclusion that traps or chemical control may be needed for the control is OK, but there are lots of other options that could be explored.

Finally, I think a very careful revision of the terminology used and English language should be checked. For instance, “Pest” should be used instead of “Plague”. I also suggest that pest eradication is never really the goal – but rather population regulation, and maintaining pests below some economic damage threshold. I was surprised that for a biological control paper / model that the authors did not use more of these terms to describe the population level required by the farm or plantation for a successful outcome.

Specific Comments:

Line 3. Suggest “biological control agents” instead of “controllers”

Line 5. Change “Considering the foregoing, the study describes” to “This study considers…”

Line 12. Change “plague” to “pest” (this should be changed here, and everywhere in the text that plague is used. The plague is generally used to refer to the Bubonic Plague or similar. Moreover, eradication of a pest is likely not possible, but regulation of its population (or for any pest) is usually the goal.

Line 19. I would not call the CBB a “natural enemy” of coffee. Natural enemy is usually referring to predators or parasitoids that control pests. Here, the CBB is a pest. So I would call this a primary coffee herbivore or a primary coffee pest.

Line 27-36 – These phrases used to describe the different control methods have valuable information, but these are incomplete sentences. Either these should be included as part of one sentence and indicated by (1) Chemical control, (2) Cultural control, etc. Or the authors should rephrase these phrases to be complete sentences. “Chemical control is a control method that consists of using insecticides….”

Line 48 – this is also an incomplete sentence: “While ants from the species Crematogaster sp. contribute to biological control by consuming immature CBBs inside the coffee beans [10].” Maybe change “while” to “In addition…”

Line 53 – change “according with” to “according to”

Line 38-59 – I would suggest to rewrite this as one paragraph all about the biological control provided by ants.

Line 60-76 – For me, this section is lacking some specific research questions or goals. Currently, this section is a listing of the different things that were investigated, more like a methods section and less like a section that outlines the different hypotheses. For instance, is the goal to look at what number of ants is necessary to regulate the CBB population using these models, and if so, which parameters are adjusted? Or is the goal to look at which life stage is most suspectable to ants, or whether a particular life stage could be better targeted by ants or chemical control in order to regulate the CBB population? I would suggest revising this section so that there are specific research goals or aims or questions that the authors want to address.

Line 65-66 – This sentence is unclear to me.

Line 83-85- I am surprised that the model did not attempt to examine differences in ant predation at different life stages, given the information shared in the introduction and the importance of different ant species at different life stages (preying on CBB within the fruit, preventing the CBB from perforating the fruits, preying on the CBB on the ground, etc.). Thus ants may be affecting both immatures and adults as well as males and females within the fruit, but there is no distinction between their effects at different life stages?

Line 150 – if so much of the ant diet is other things besides the CBB, is that reflected in the equations? It appears that most of the ant population growth and mortality and carrying capacity is more related to the CBB than alternative prey. Or are the alternative prey ignored in this case?

In the methods section generally, it is not clear how the parameter values were selected and where the data comes from for the starting values for each of the parameters included in the model.

Line 359 – please translate the subheader to English

Line 373 – but hypothetically assigned based on what assumption? Why that value?

Line 379 – Is there any citation or information for the population of ants? A single nest of Wasmannia might contain thousands of individuals, and a nest of Solenopsis might contain thousands of individuals. So at the level of a plantation, even a small one with 10 nests of those organisms might have tens of thousands of individuals, which is a population that could be quite large. If the authors are arguing that the model is considering one ant species, it might be good to parametrize the model based on one particular species and its population size and any relevant references.

Line 404 – but if the models only include one species of ant and its population as moderating the population of the CBB, why cannot a second or third or fourth species of ant contribute? What about predation by other organisms like birds? What about parasitism? Why must insecticides or alcohol traps be the secondary mortality factor in order to regulate the population of the CBB?

Line 418 – need to revise “[? ].”

The English language content should be carefully checked. There are many words that are not properly translated for biological control literature, and there are many incomplete sentences that make it difficult to understand the manuscript.

Reviewer 2 Report

Please see the attached report

The description of the species in the introduction will benefit from the inputs (language) of an entomologist familiar with them. 

Round 2

Reviewer 1 Report

Review for Insects 2375699_R1

Mathematical Model for the Biological Control of the Coffee Berry Borer Hypothenemus hampei through Ant Predation

General Comments:

I appreciate the authors work in completing the revisions for this manuscript.

I still have some significant concerns about the understandability for a general audience.

I have made several suggestions below about ways that I think can improve the flow and readability of the manuscript. These include adding a more concise description of the objectives, including biological terms (and not just mathematical terms) in the description of the propositions.

I am also uncertain why the descriptions of the parameter values are not described in the methods, and why when some of them are taken from the literature, those papers are not cited and the values are not discussed.

I have a very difficult time understanding this manuscript and the results, and whether the models reveal the conclusions that the authors make. Some of this is because of the way the authors describe their methods and the different modeling approaches. I believe that by shifting some of the methods description, and descriptions of the results, the paper will be more understandable by a wider audience, including people, like myself, who are very interested in biological control of the CBB.

Specific Comments:

There are not tracked changes in the manuscript, which is making it really difficult to see what the authors have changed in their submission from the previous version.

Line 20-22 – Does this mean that both of the conditions of co-existence (maybe instead call population persistence of both the predator and prey) as well as pest eradication were determined?

Line 78-79 – Something about this part, “feed on the CBB prey on this pest” is awkward and should be rewritten.

Line 88 – change to “according to”

Line 90-94 – This is a run on sentence, and is difficult to understand. Given that this is a key sentence in describing the purpose of the study, it is really important that this be extremely clear.

Line 85-100 – I appreciate that the authors have made efforts to revise this section of the introduction to make the study objectives clearer. That said, it still is not very clear to me what specific research questions are addressed. I think that I understand that the authors want to understand the impacts of ants and other control measures on the population persistence of the CBB, and that they are also looking at the impacts of ants and other control measures on the eradication (at least theoretical) of the CBB population. I am not sure if co-existence is the best way to describe this, because it is not clear if population persistence of the ant is a goal of the research or not? I would suggest to try to clarify this further. I think that putting in specific questions would be really helpful such as (Q1: What are the conditions under which the CBB population might persist with ant biological control and other control methods? Q2: What are the conditions under which the CBB population might be eradicated by ant biological control combined with other control methods?). It should also be clear what are those other control methods here.

Line 97 – I would suggest that the portion starting “unlike works such as…” become a different sentence. Distinguishing your study from previous works is important, but it should not be presented in the same sentence as the study questions or objectives. You can instead say, “Our study is novel in that we introduce both ant biological control and other mortality factors for CBB, unlike [] and [] that only studied one mortality factor at a time”

Line 140 – change “Thirdly” to “Third”

Line 399 – which of the values were taken from the literature, and from which publications? Maybe that info could be added into the Table 1.

3. Results – In the results section, I am very confused by the different propositions. I understand that they way that they are worded may be standard in the theoretical ecology or biological control literature. However, as a scholar of CBB and coffee agroecosystems, I would like to understand what each proposition means.  This is just an example (with incorrect terms, certainly) about how the authors might address this point. The current text describing proposition 2 reads “Proposition 2. The stability of equilibrium points E1, E2, E3 and E4 of the system (4) depends on thresholds B0 and Y0 given in (8) and (9), respectively, in the following manner:”. Is there a way to describe this in ecological terms? Yes, the different parameters have been described elsewhere, but there are so many it is easy to lose track and to mentally substitute the processes and population sizes therein. So perhaps saying something like, “Proposition 2. The stability of the equilibrium points depends on the population size of the immature CBB and on the attack rate by the ants, as well as chemical application rates”. I am certain that this is not an accurate description of the authors statement of Proposition 2, I am merely using my description to illustrate what a more ecological / biological description would look like. I think having those statements would dramatically improve the understandability of the manuscript.

Line 459-456 – This is a helpful explanation. The sentence is long, however, so I suggest splitting into two.

The English for the article is OK, but there is a lot of room for improvement. I think in addition, the mathematical terminology needs to be better explained for a general audience; this may include adding in more biological terms in the description of the different model propositions.

Author Response

Consulte el archivo adjunto

Reviewer 2 Report

This is a good improvement of the original manuscript. However,  in Section 2 Material and methods item 3 including references is a good response, a clear and detailed description of the formula was expected. 

The narrative of this manuscript can be much improved if it were reviewed by a professional in the English language. 

Author Response

We thank the reviewer for their valuable corrections, which have been crucial in the writing of this new version of the article, which is now easier to understand.

Round 3

Reviewer 1 Report

Review for Insects 2375699 version3

I quickly reviewed this 3rd version of the manuscript. While I still have some problems to understand the paper, I think that the authors have added sufficient explanations that it is more understandable now. I thank the authors for their attention to the revisions.